# Ancestral L-amino acid oxidases for deracemization and stereoinversion of amino acids

Shogo Nakano [1,2✉], Kohei Kozuka[1,2], Yuki Minamino[1], Hiroka Karasuda[1], Fumihito Hasebe [1] & Sohei Ito [1]

L-amino acid oxidases (LAAOs) can be applied to convert racemic amino acids to D-isomers, which are potential precursors of pharmaceuticals. However, this application is hampered by the lack of available stable and structure-determined LAAOs. In this study, we attempt to address this limitation by utilizing two ancestral LAAOs: AncLAAO-N4 and AncLAAO-N5. AncLAAO-N4 has the highest thermal and temporal stabilities among the designed LAAOs that can be used for deracemization and stereoinversion. AncLAAO-N5 can provide X-ray crystal structures, which are helpful to reveal substrate recognition and reaction mechanisms of LAAOs at the molecular level. Next, we attempted to improve activity of AncLAAO-N4 toward L-Val through a semi-rational protein engineering method. Three variants with enhanced activity toward L-Val were obtained. Taken together, we believe that the activity and substrate selectivity of AncLAAOs give them the potential to be key enzymes in various chemoenzymatic reactions.

[1] Graduate School of Integrated Pharmaceutical and Nutritional Sciences, University of Shizuoka, 52-1 Yada, Suruga-ku, Shizuoka 422-8526, Japan. [2]These authors contributed equally: Shogo Nakano, Kohei Koduka. ✉email: snakano@u-shizuoka-ken.ac.jp

Enantiopure amino acids are attractive compounds which have the potential to be utilized in the synthesis of chemicals such as pesticides, antibiotics, and pharmaceuticals[1]. Proteinogenic amino acids have been produced on a large scale using a fermentation approach, whereas synthesis of non-proteinogenic amino acids, such as enantiopure amino acid derivatives, remains a challenging task[2]. Many attempted approaches to solve this challenge have been reported and all involve synthesizing chiral amino acids[3–11]; in this study, we focused on the deracemization approach utilizing amino acid oxidases and chemical reductants[12,13]. The deracemization was performed by the following steps repeatedly; D- or L-amino acids in racemates were oxidized to α-imino acids by amino acid oxidases, and the resulting imino acids were reduced to racemates by chemical reductants (Fig. 1). Chiral L-amino acids could be synthesized utilizing D-amino acid oxidase (DAAO) bearing broad substrate selectivity[14]. Enzyme functions of DAAO were elucidated at the molecular level through multiple X-ray crystal structures[15,16] and biochemical analysis[17,18]. These findings contribute to the design of DAAO variants bearing new substrate selectivity and high activity[14,19]. High-throughput screening of the variants generated by a protein engineering method, such as directed evolution and rational design, is possible by detecting $H_2O_2$ production with colorimetric methods[19,20]. Undoubtedly, this method enables us to efficiently select highly functional variants from a large mutation library. In fact, many of the DAAO variants were designed and adopted for the purpose of obtaining enantiopure L-amino acids[14,21].

Theoretically, chiral D-amino acids would be synthesized utilizing L-amino acid oxidase (LAAO) as an alternative to DAAO, due to its broad substrate selectivity. However, the application of LAAOs to deracemization is difficult for several reasons. A key challenge is low productivity of LAAOs by heterologous expression. This low productivity may be due to high toxicity of the enzymes in the host cells[22]. L-amino acid deaminase (LAAD) is utilized as an alternative to LAAO[3,11,23], and is a FAD-dependent enzyme which catalyzes the deamination of L-amino acids[24–26]. LAAD generates little $H_2O_2$ during the reaction which is a key factor to improving productivity and the ability to synthesize α-keto acids from L-amino acids[24]. On the other hand, LAAD requires an external electron transporter system to reoxidize $FADH_2$ generated by the reaction[24,25]. LAAOs do not require this system because the enzymes complete cofactor regeneration utilizing $O_2$ in the solution; this is one of the advantages of using LAAOs for bioconversion reactions.

Based on this knowledge, LAAO has remained an attractive target to redesign enzymatic properties by protein engineering

methods. Recently, we designed an artificial LAAO (AncLAAO-N1 in this study, and AncLAAO in a previous study[27]) bearing broad substrate selectivity (>10 L-amino acids) and high productivity in an *Escherichia coli* expression system using ancestral sequence reconstruction (ASR). This enzyme can be applied to the deracemization of aromatic D-amino acids[27]. Simultaneously, several barriers must be overcome prior to enabling AncLAAO-N1 to deracemize various compounds as well as DAAO. Low thermal stability and lack of experimental data for structural and functional analysis of AncLAAOs would prevent this application.

In this study, we attempted to redesign another AncLAAO bearing higher thermal and temporal stability than AncLAAO-N1 by restoring ancestral sequences on every node of the phylogenetic tree. From the restored sequences, two AncLAAOs (AncLAAO-N4 and AncLAAO-N5) were selected as samples which satisfied this purpose. The substrate recognition mechanism of AncLAAO was predicted through crystal structures of AncLAAO-N5 determined by the iodide-single wavelength anomalous dispersion (iodide-SAD) method. With reference to this mechanism, we attempted to design AncLAAO-N4 variants which improve activity toward L-Val by a protein engineering method.

## Results

**Acquisition of highly stable artificial LAAOs through comprehensive design by ASR.** AncLAAO-N1 is one of the previously designed AncLAAOs generated by the ASR method utilizing six LAAO candidates from *Pseudoalteromonas*[27]. This suggested that there are a total five ancestral LAAOs (AncLAAOs) including AncLAAO-N1, and the enzymatic properties of the remaining four AncLAAOs have not been characterized. In the AncLAAOs, there are new enzymes which overcome functional drawbacks of AncLAAO-N1, such as low thermal and temporal stability (black-filled square in Fig. 2a, b, respectively). Specifically, the long-term stability of AncLAAO-N1 must be improved to reduce the amount of the enzyme required to perform the deracemization. The activity of the enzyme rapidly decreased to less than 15% when the sample was incubated for 60 min at 30 °C (black-filled square in Fig. 2b).

To find new AncLAAOs which overcome these disadvantages, we attempted to express a total of five AncLAAOs (AncLAAO-N1, AncLAAO-N2, AncLAAO-N3, AncLAAO-N4, and AncLAAO-N5) which were generated by the ASR method in a previous study (Supplementary Data 1 for protein sequences and Supplementary Data 2 for nucleotide sequences)[27]. Here, we successfully expressed AncLAAO-N1, which is called AncLAAO in the previous study[27], and attempted to do the same for the four

**Fig. 1 Deracemization reaction cascade to obtain enantiomerically pure D-1.** In the reaction, AncLAAO-N4 oxidizes L-amino acids to imino acids, and the product is reduced by chemical reductant, $NH_3:BH_3$.

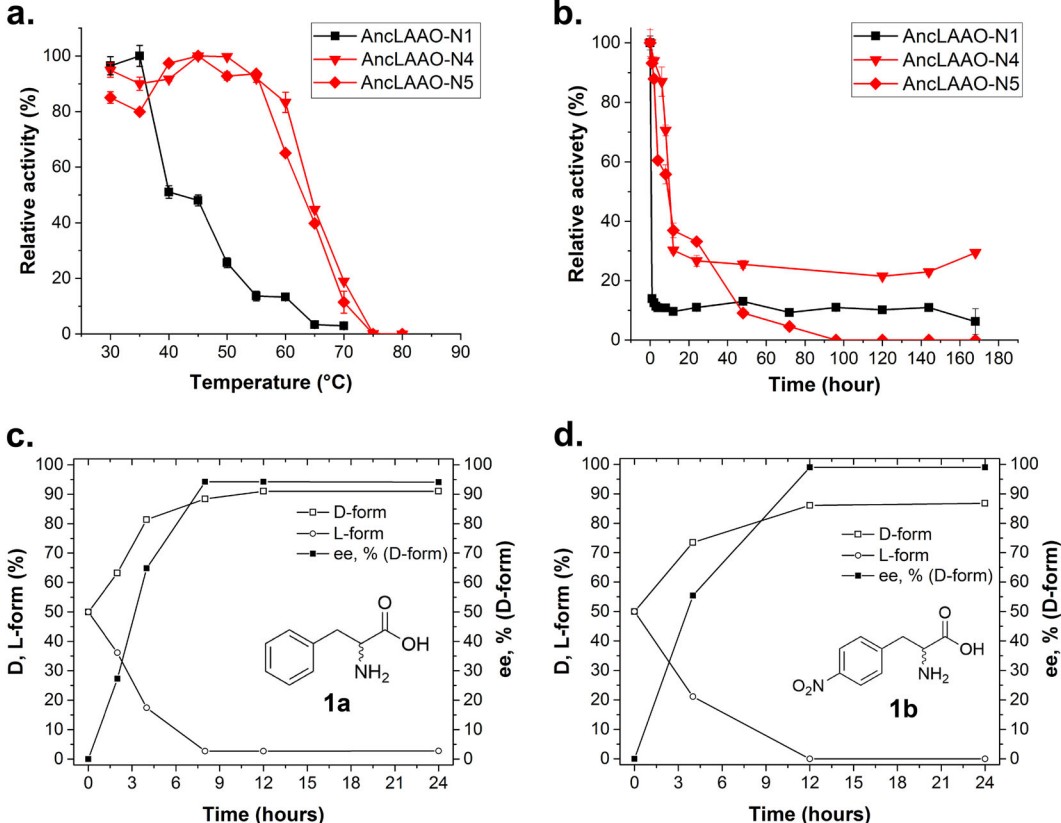

**Fig. 2 Enzyme functional analysis of AncLAAOs.** Evaluation of short-term (**a**) and long-term (**b**) thermal stability of designed AncLAAOs. In (**a**), activity was measured after incubation of the samples for 10 min at different temperatures. In (**b**), activity was measured after incubation of the samples at 30 °C for different times (0–168 h). Relative activity was calculated by interpreting the highest activity of each sample as 100%. Relative activity changes of AncLAAO-N1, AncLAAO-N4, and AncLAAO-N5 with dependence on temperature (a) and time (b) were represented as filled square, inverted triangle, and diamond, respectively. A total of 1.1, 0.14, and 0.38 µg of AncLAAO-N1, N4, and N5 were utilized in the measurement. Time course of D-1a (c) and D-1b (d) production from racemates by deracemization reaction. Chiral HPLC chromatogram before and after the reaction is represented in Supplementary Fig. 4.

remaining AncLAAOs. Multiple sequence alignment and phylogenetic tree analysis of AncLAAOs are shown in Supplementary Figs. 1 and 2, respectively. These results indicate that sequence identity among AncLAAOs are quite high (73–99.5% shared identity, Supplementary Table 1) and they should have LAAO activity similar to AncLAAO-N1[27]. To confirm this, the five AncLAAOs were expressed to estimate their activity. Although all of the AncLAAOs had activity toward L-Met in initial screening, as expected, the enzymatic properties of AncLAAO-N2 and AncLAAO-N3 could not be evaluated accurately due to their low stability and detachment of FAD, among other reasons. Thus, only AncLAAO-N4 and AncLAAO-N5 were adopted as research subjects in this study. Overexpression of AncLAAO-N4 and AncLAAO-N5 can be achieved utilizing the BL21(DE3) expression system; more than 200 units/L of the samples can be obtained after purification using Ni-affinity chromatography (Supplementary Table 2). The thermal stability of AncLAAO-N4 and AncLAAO-N5 was clearly improved over that of AncLAAO-N1; the $t_{1/2}$ values of AncLAAO-N4 and AncLAAO-N5 were approximately 25 °C higher than that of AncLAAO-N1 (red lines in Fig. 2a). Optimal pH values of AncLAAO-N1, N4, and N5 were 7.0, 7.5, and 6.0, respectively (Supplementary Fig. 3b). AncLAAO-N4 exhibits strong activity over a broad range of pH; residual activity was >80% from 6.5 to 8.0 (Supplementary Fig. 3b). Next, we evaluated the long-term stability of the AncLAAOs. AncLAAO-N5 completely lost its activity during the incubation (Fig. 2b, red diamond). In addition, analysis of enzymatic properties for AncLAAO-N5 is difficult because its

activity was clearly and immediately reduced by diluting the sample. Only AncLAAO-N4 improved the stability; AncLAAO-N4 retained approximately 30% of its activity after incubation for 1 week at 30 °C (red line in Fig. 2b). There was no activity loss for AncLAAO-N4 by the dilution, and we can easily confirm the reproducibility of the analysis. In addition, AncLAAO-N4 had broad substrate selectivity (13 L-amino acids) similar to that of AncLAAO-N1 (Supplementary Table 3). The optimal temperature for AncLAAO-N4 was 50 °C, but the residual activity was >55% between 20 and 65 °C (Supplementary Fig. 3c).

**Deracemization and stereoinversion to D-amino acid derivatives by AncLAAOs.** Based on these results, AncLAAO-N4 has the potential to be an alternative to AncLAAO-N1 for deracemization. To confirm this point, we performed deracemization for several phenylalanine derivatives at preparative scale utilizing AncLAAO-N4 (Supplementary Figs. 4 and 5). Deracemization of D,L-Phe (D,L-**1a**) and 4-nitro-D,L-Phe (D,L-**1b**) was performed using a combination of AncLAAO-N4 and chemical reductant, $NH_3:BH_3$ (Fig. 1). First, the optimal temperature was determined to perform the deracemization reaction utilizing D,L-Phe as substrates. The reactions progressed smoothly at 20 °C (Fig. 2c) and 30 °C (Supplementary Fig. 3d) but not at 40 °C (Supplementary Fig. 3e). The best *ee* value could be obtained at 20 °C (94% [*ee*], Table 1) compared to 30 °C (84% [*ee*], Supplementary Fig. 3d) and 40 °C (28% [*ee*], Supplementary Fig. 3e). Thus, we performed the deracemization reaction at 20 °C. The reaction was completed within 9 h (D,L-**1a**, Fig. 2c) and 12 h (D,L-**1b**, Fig. 2d),

**Table 1 Deracemization and stereoinversion of racemic phenylalanine derivatives (D,L-1a-e) and L-amino acids (L-1f-h) to D-amino acid derivatives (D-1a–h).**

| Substrate | R | Specific activity (U/mg)[a] | Conversion (%)[b] | ee (%)[c] | Reaction time/amounts of substrates |
|---|---|---|---|---|---|
| Deracemization | | | | | |
| D,L-1a | Ph | 7.6 ± 0.1 | 91 | 94 | 12 h/166 mg |
| D,L-1b[d] | $4-NO_2-Ph$ | 12.8 ± 0.2 | 87 | >99 | 12 h/210 mg |
| D,L-1c | 4-OMe-Ph | 10.2 ± 0.1 | 87 | >99 | 24 h/195 mg |
| D,L-1d | $4-NH_2-Ph$ | 0.38 ± 0.01 | 95 | 97 | 48 h/180 mg |
| D,L-1e[d] | 3-F-Ph | 9.7 ± 0.1 | 84 | >99 | 24 h/183 mg |
| Stereoinversion | | | | | |
| L-1f | Gln | 27.9 ± 1.1 | 98 | >99 | 24 h/146 mg |
| L-1g | Glu | 14.8 ± 0.2 | 80 | >99 | 72 h/147 mg |
| L-1h | Trp | 9.5 ± 0.2 | 80 | >99 | 72 h/204 mg |

[a]The specific activity was estimated utilizing 5 mM of D,L- and L-amino acid derivatives as substrates.
[b]The conversion rate was estimated from peak area obtained by HPLC analysis (Supplementary Figs. 4 and 5 for the deracemization and Supplementary Fig. 6 for the stereoinversion). The conversion was performed under preparative scale. For the conversion, a total of 3 mg (1a and 1b) and 2 mg (1c–1h) of purified AncLAAO-N4 were utilized.
[c]ee was calculated with reference to the equation described in the Methods section.
[d]Isolated yield was estimated as follows: 92%[1b] and 86%[1e].

and highly enantiopure D-phenylalanine derivatives (94% [ee] for D-1a and >99% [ee] for D-1b, Supplementary Fig. 4) could be obtained (Table 1). Stereoinversion of three L-amino acids (L-1f, 1g, and 1h) to D-isomers could be achieved with high conversion rate (>80%) and ee rate (>99%) (Supplementary Fig. 6 and Table 1), also indicating that AncLAAO-N4 is a strong biocatalyst candidate to synthesize enantiopure D-amino acid derivatives. Deracemization was performed for the other three racemic phenylalanine derivatives (D,L-1c, D,L-1d, and D,L-1e; Table 1). Finally, three of the five racemates converted to D-enantiomer completely (>99% ee for 1b, 1c, and 1e), and the remaining two racemates were mostly converted to D-enantiomer at preparative scale (94% ee for 1a, and 97% ee for 1d) (Supplementary Figs. 4 and 5; Table 1). Furthermore, by utilizing AncLAAO-N4, the necessary amount of enzymes could be reduced to perform the deracemization; a total of 3 mg (1a and 1b) and 2 mg (1c-h) of AncLAAO-N4 was used in the reaction (Supplementary Figs. 4–6), which is less than a half of the amount of AncLAAO-N1 (7 mg[27]) required under similar conditions to achieve an identical amount of deracemized racemic amino acids.

**Overall structure of AncLAAO-N5.** By comprehensive design and functional analysis of AncLAAOs, we obtained AncLAAO-N4, which has the highest stability among the AncLAAOs and can be applied to deracemizations. The next challenge is to change the substrate selectivity of AncLAAO-N4 by protein engineering methods. However, in the current situation, the conversion is difficult because the molecular mechanism which determines how AncLAAOs recognize a broad range of L-amino acids is unknown. Structural information of AncLAAOs is helpful to predict the mechanism, however, no crystal structures of AncLAAOs are available. In fact, for the most similar structure to AncLAAO-N1, which registered in PDB (Tryptophan 2-monooxygenase from Pseudomonas savastanoi, PDB ID: 4IV9[28]), the query cover rate was only 10% and the E value was 0.001. This also suggested that the construction of a homologous model of AncLAAOs is difficult at this time. Determination of crystal structures of AncLAAOs is the only way to elucidate the mechanism at the molecular level.

To tackle with this challenge, we first performed screening of crystallization conditions for all of the AncLAAOs. We found that only AncLAAO-N5 generated crystals for which the resolution was high enough to represent electron density map of substrates. Utilizing these crystals, the X-ray crystal structure of a ligand-free form of AncLAAO-N5 was determined at a 2.4 Å resolution with the iodide-SAD phase determination method. Crystallographic parameters were indicated in Table 2. The overall structure of AncLAAO-N5 is shown in Fig. 3a, indicating that there is one protein molecule in the asymmetric unit. On the other hand, PISA analysis suggested that AncLAAO-N5 forms a dimer in the crystals; interface area and CCS scores were 1910 Å$^2$ and 0.432, respectively[29]. Furthermore, previous studies indicated that other LAAOs have a dimer form[30,31]. AncLAAO-N5 was crystallized utilizing concentrated samples which were collected at the dimer fraction of gel-filtration chromatography (Supplementary Fig. 7a, b). Thus, we predicted that AncLAAO-N5 forms a dimer as shown in Fig. 3a. From the electron density map, FAD is bound to AncLAAO-N5 as expected.

Next, the AncLAAO-N5 structure was analyzed using the DALI server[32]. The result suggested that AncLAAO-N5 appears to belong to the flavin-dependent amine oxidase (FAO) superfamily[33], like LAAO from Calloselasma rhodostoma (CrLAAO) and L-lysine oxidase from Hypocrea rufa (Supplementary Table 5). The superimposed structures of AncLAAO-N5 and CrLAAO indicated that they share a similar fold with each other (Supplementary Fig. 8b) but have unique active-site structures (Supplementary Fig. 8c). As well as CrLAAO, other proteins belonging to the FAO superfamily also have structural similarity to AncLAAO-N5 as the RMSD values were distributed from 2.4 to 3.3 Å in spite of the low (less than 20%) sequence identity between them (Supplementary Table 5).

**Substrate recognition and reaction mechanism of AncLAAOs predicted from crystal structures of AncLAAO-N5.** AncLAAOs can recruit a broad range of substrates into the active site via the substrate entrance pathway. Here, we predicted the pathway by searching tunnels of the AncLAAO-N5 structure which connected the active site to the entrance using the CAVER software[34] (Fig. 3b). The tunnel is mainly formed by hydrophobic residues (K160, L229, F231, L247, F251, F391, Y447, and W571; Fig. 3b); therefore, this may make it convenient to introduce hydrophobic L-amino acids into the active site.

Finally, we attempted to predict the substrate recognition mechanism of AncLAAOs at the molecular level. By soaking AncLAAO-N5 crystals in cryoprotectant solution containing substrates, crystal structures of L-Gln, L-Phe, and L-Trp binding forms of AncLAAO-N5 were obtained in 2.6, 2.4, and 2.2 Å resolution, respectively (Table 2). Here, AncLAAO-N5 crystal was bleached by soaking, suggesting that FAD in the crystal was completely reduced to FADH. In this situation, AncLAAO-N5 should lose enzymatic activity. There is a large amount of substrate compared with AncLAAO-N5, and therefore, the

**Table 2 Statistics of X-ray diffraction data collection of AncLAAO-N5 for native and substrate binding forms (L-Gln, L-Trp, and L-Phe binding form).**

|  | Native | Iodide-SAD | L-Gln binding | L-Trp binding | L-Phe binding |
|---|---|---|---|---|---|
| Space group | I422 | I422 | I422 | I422 | I422 |
| Unit cell parameters |  |  |  |  |  |
| a (Å) | 131.9 | 132.6 | 132.6 | 132.5 | 132.0 |
| b (Å) | 131.9 | 132.6 | 132.6 | 132.5 | 132.0 |
| c (Å) | 191.2 | 191.9 | 191.8 | 191.7 | 191.4 |
| α (degree) | 90.0 | 90.0 | 90.0 | 90.0 | 90.0 |
| β (degree) | 90.0 | 90.0 | 90.0 | 90.0 | 90.0 |
| γ (degree) | 90.0 | 90.0 | 90.0 | 90.0 | 90.0 |
| X-ray source | BL5A (PF) | BL5A (PF) | BL5A (PF) | BL5A (PF) | BL5A (PF) |
| Wavelength (Å) | 1.00 | 1.70 | 1.00 | 1.00 | 1.00 |
| Resolution (Å) | 46.6-2.4 (2.49-2.40) | 48.0-2.6 (2.74-2.60) | 48.0-2.6 (2.74-2.60) | 47.9-2.4 (2.53-2.4) | 47.8-2.2 (2.32-2.20) |
| No. of reflections[a] | 877,891 | 2,780,097 | 704,737 | 899,166 | 1,156,916 |
| No. of unique reflections | 33,315 | 26,611 | 26,591 | 33,554 | 43,030 |
| Completeness (%) | 100 (99.9) | 100 (100) | 100 (99.9) | 100 (99.9) | 100 (99.8) |
| I/sig(I) | 25.9 (5.5) | 58.9 (16.0) | 26.5 (6.2) | 30.8 (6.6) | 28.9 (7.0) |
| $CC_{1/2}$ | 0.999 (0.967) | 1.00 (0.998) | 1.00 (0.953) | 1.00 (0.966) | 1.00 (0.978) |
| $R_{merge}$[b] | 0.125 (0.674) | 0.085 (0.380) | 0.119 (0.687) | 0.107 (0.596) | 0.107 (0.597) |
| B of Wilson plot (Å)$^2$ | 29.5 | 41.8 | 39.7 | 30.8 | 24.2 |
| Iodide sites |  | 39 |  |  |  |
| FOM before DM[c] |  | 0.38 |  |  |  |
| FOM after DM[c] |  | 0.69 |  |  |  |
| R[d] | 0.171 |  | 0.207 | 0.208 | 0.173 |
| $R_{free}$[e] | 0.231 |  | 0.276 | 0.270 | 0.211 |
| RMSD of geometry |  |  |  |  |  |
| Bond length (Å) | 0.008 |  | 0.009 | 0.008 | 0.008 |
| Bond angle (degree) | 0.92 |  | 1.04 | 0.92 | 0.88 |
| Geometry |  |  |  |  |  |
| Ramachandran outlier (%) | 0.5 |  | 0.7 | 0.3 | 0.3 |
| Ramachandran favored (%) | 99.5 |  | 99.3 | 99.7 | 99.7 |
| Average B factor (Å)$^2$ |  |  |  |  |  |
| Protein atoms | 37.9 |  | 44.1 | 37.8 | 34.4 |
| Ligand atoms | 30.8 |  | 41.3 | 33.3 | 29.5 |
| Solvent atoms | 35.7 |  | 39.6 | 36.3 | 35.4 |
| PDB code | 7C4K |  | 7C4L | 7C4M | 7C4N |

[a] Sigma cutoff was set to none ($F > 0\sigma F$).
[b] $R_{merge} = \Sigma_h \Sigma_i |I_i(h) - \langle I(h) \rangle| / \Sigma_h I(h)$, where $I_i(h)$ is the $i^{th}$ measurement of reflection $h$, and $\langle I(h) \rangle$ is the mean value of the symmetry-related reflection intensities. Values in brackets are for the shell of the highest resolution.
[c] FOM before/after DM means that the figure of merit before/after density modification.
[d] $R = \Sigma ||F_o| - |F_c|| / \Sigma |F_o|$, where $F_o$ and $F_c$ are the observed and calculated structure factors used in the refinement, respectively.
[e] $R_{free}$ is the R-factor calculated using 5% of the reflections chosen at random and omitted from the refinement.
[f] n.d., not determined.

substrate could bind to the active site as well and aid in the structure determination of the substrate binding form of R-amine oxidase[19,35]. The electron density map at the active site of the L-Gln (green, Fig. 3c), L-Phe (cyan, Fig. 3c), and L-Trp binding forms (magenta, Fig. 3c) indicates that the interaction formed between the main chain of the substrates and active site residues in AncLAAO-N5 involves the same amino groups regardless of the substrate; the main chain carboxyl and amino groups of the substrates form hydrogen bonds with R89 and G570, respectively (dotted line in Fig. 3c). Here, there are many factors why complete conversion of several racemic amino acids (such as D,L-**1a** and D,L-**1d**; Table 1) to D-isomers could not be achieved, including unsuitability of enzymatic properties of AncLAAO-N4 exemplified by catalytic activity, substrate-affinity hydrolysis, and reduction of imino acids. Related to these factors, reaction products, keto acids, and carboxylic acids may be inhibitors of AncLAAOs as well as other LAAOs[36]. On the other hand, the interaction formed between the side chain of the substrates and the active site residues is somewhat different; L-Gln formed a hydrogen bond with Q537, while L-Phe and L-Trp formed

hydrophobic interactions with several active site residues, such as F231 and V535. Here, Q537 is the only residue which has a side chain amide group placed to form an interaction with the substrate, suggesting that the residue is important for recognizing L-Gln and L-Glu as substrates (Fig. 3c). F231 is the only residue of which conformation is dynamically changed with dependence on the substrates (Fig. 3c). This suggested that F231 may be one of the important residues that enables AncLAAOs to recognize a broad range of substrates.

Summarizing the results, we predicted a molecular mechanism that enables AncLAAOs to recognize and oxidize substrates utilizing L-Phe and D-Phe as model substrates (Fig. 4). Here, the oxidation of L-Phe could progress with the same mechanism as indicated by other LAAOs[22,30,31], as the binding and interaction modes between the main chain of L-Phe and active site residues in AncLAAO-N5 are identical to other LAAOs. For example, four residues in AncLAAO-N5 (R89, Y447, G570, and W571 in Supplementary Fig. 8c, cyan) that formed an interaction with the main chain of L-Phe are conserved in CrLAAO (R90, Y372, G464, and W465 in Supplementary Fig. 8c, orange). As for

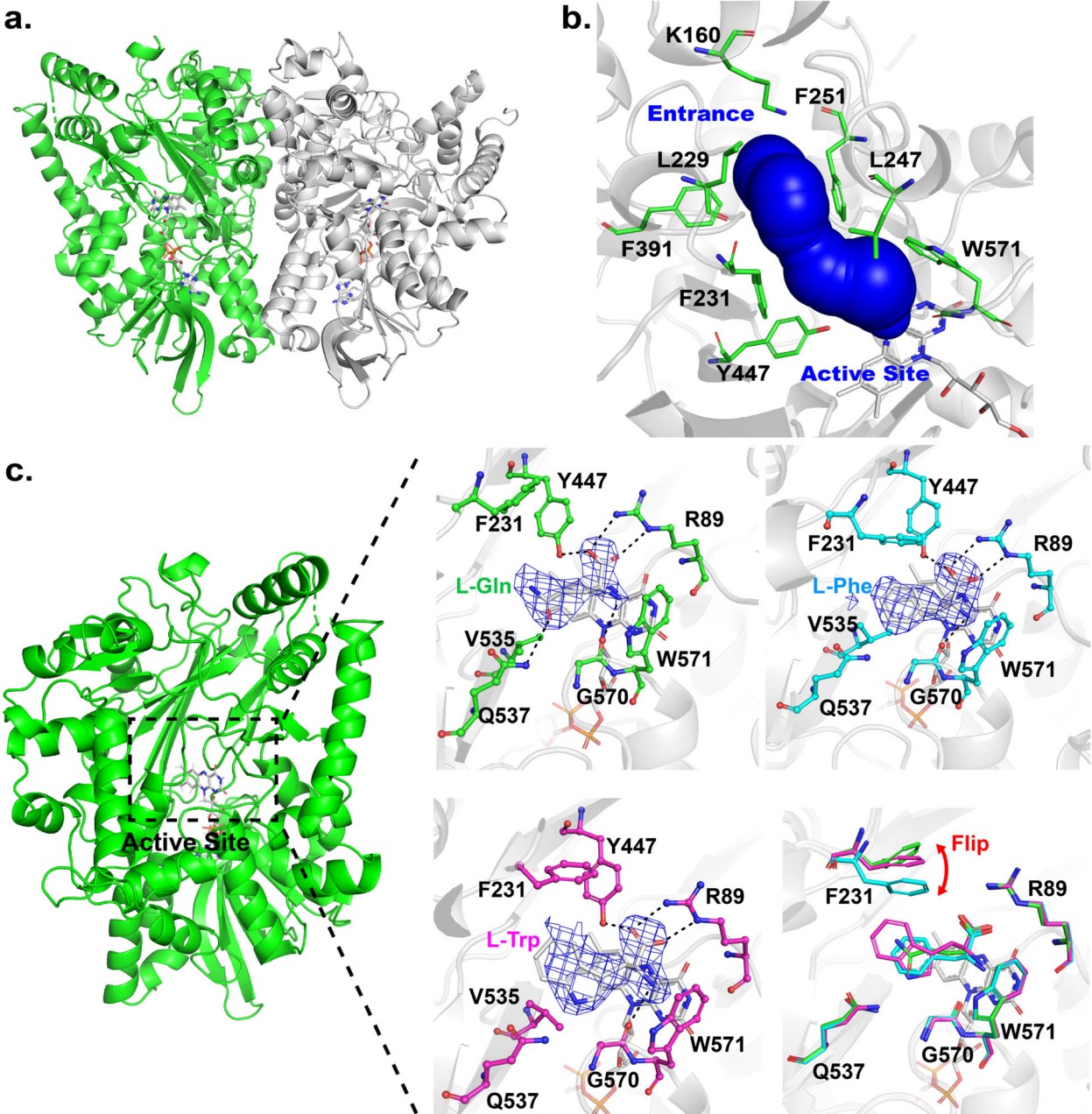

**Fig. 3 X-ray crystal structure analysis of AncLAAO-N5.** Structure of ligand-free form of AncLAAO-N5 (a). No ligand binds to the active site according to the polder $F_o$-$F_c$ omit map (Supplementary Fig. 8a). Predicted substrate entrance tunnel of AncLAAO-N5 (b). The tunnel was assigned utilizing the Caver 3.0 software. The tunnel was formed by the following eight residues: K160, L229, F231, L247, F251, F391, Y447, and W571. Active site structures of the ligand binding form of AncLAAO-N5 (c). L-Gln, L-Phe, and L-Trp binding forms were represented as green, cyan, and magenta, respectively. The polder $F_o$-$F_c$ omit maps of L-Gln, L-Phe, and L-Trp were contoured at 5.0, 5.0, and 4.0 σ, respectively. The superimposed structures of the ligand binding forms suggested that conformational change of F231 is induced by binding of the ligands.

oxidation of the amino group of L-Phe, a solvent water molecule catalyzes the deprotonation reaction (Fig. 4). Simultaneously, a hydride on the Cα atom of L-Phe is transferred to the N5 atom of FAD (Fig. 4). For D-Phe, the deprotonation reaction could not be performed because the proton is remote from the N5 atom of FAD (Fig. 4). On the other hand, the interaction between the side chain of L-Phe and active site residues are unique in LAAOs. In AncLAAO-N5, four residues (F231, L247, V535, and Q537) can form interactions with the side chain of L-Phe (Fig. 4). By mutating these residues, we may be able to alter the substrate selectivity of AncLAAOs.

**Enzyme kinetics analysis for loss-of-function variants of AncLAAO-N4.** Crystal structures of AncLAAO-N5 enables us to predict the substrate recognition and reaction mechanism of AncLAAOs (Fig. 4). Based on the results, we can investigate the following two subjects: 1. enzyme functional analysis of loss-of-function variants of AncLAAOs to validate the predicted mechanism; 2. generation of gain-of-function variants of AncLAAOs by protein engineering method. In this section, we attempted to obtain these variants by mutating AncLAAO-N4 instead of AncLAAO-N5; the highly stable AncLAAO-N4 is likely to bear negative effects brought by the mutation more than

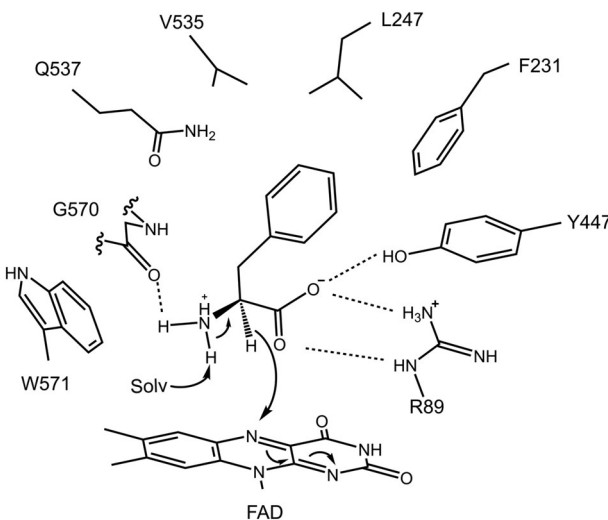

**Fig. 4 Proposed substrate recognition and reaction mechanism of AncLAAO.** The reaction mechanism was proposed with reference to previously reported LAAOs.

AncLAAO-N5. Active site structures between AncLAAO-N4 and AncLAAO-N5 are identical to each other (arrow head and asterisk in Supplementary Fig. 1) and they share high sequence identity (73%, Supplementary Table 1). In fact, there are a total of 66 residues around the 10 Å region of the substrate (L-Gln), and only R258 is not conserved in the five designed AncLAAOs; the main chain carbonyl group of R258 is located in this region. This suggested that, although optimal pH values are different among AncLAAO-N1, N4, and N5 (Supplementary Fig. 3b), predictions for substrate recognition and reaction mechanisms obtained by the analysis of AncLAAO-N5 structures would be helpful to determine the mechanisms of AncLAAO-N4. An important point is that AncLAAO-N4 is one residue shorter than that of AncLAAO-N5; R89 and Y447 in AncLAAO-N5 correspond to R88 and Y446, respectively, in AncLAAO-N4 (Fig. 5).

First, enzyme kinetics analysis of two AncLAAO-N4 variants, R88A and Y446F, was performed to validate the functional importance of these residues; R88 and Y446 form hydrogen bonds with the main chain of the substrate (Fig. 5a), and the mutation of these residues generated inactivated variants. Plots of initial velocity and kinetic parameters are shown in Fig. 5b and Table 3, respectively. The activity of R88A was clearly decreased; the $k_{cat}$ and $k_{cat}/K_m$ values of R88A were less than three orders of magnitude than those of AncLAAO-N4 (Table 3). This suggested that R88 of AncLAAO-N4 is an essential residue for the catalytic process to occur. Comparable $K_m$ values between AncLAAO-N4 and the R88A variant (Table 3) suggested that residues interacting with the side chain of the substrate (L-Met) is important for substrate recognition. R88 may work to orient the substrate in an appropriate position to allow the reaction to proceed. On the other hand, the decrease of the parameters of the Y446F variant was moderate; $k_{cat}$ and $k_{cat}/K_m$ values of Y446F was about 7- and 50-fold lower, respectively, than that of AncLAAO-N4. Y446 may work to place the substrate in the position optimal for allowing the reaction to proceed efficiently.

**Enzyme kinetics analysis for gain-of-function variants of AncLAAO-N4.** Next, we attempted to generate AncLAAO-N4 variants which have activity towards new substrates to prove that AncLAAO-N4 is a suitable target to obtain useful variants. In this study, L-Val was adopted as a model substrate. In the designed AncLAAOs, the relative activity toward L-Val is clearly low

(0.73%, Supplementary Table 3) compared with the activity toward L-Gln, and the activity should be improved with the application of the enzymes in the deracemization to D-Val. Structural analysis of AncLAAO-N4 suggested that steric hindrance of the indole ring (W570) prevents the placement of Cβ-branched amino acids, such as L-Ile and L-Val (Fig. 5c). W570 is broadly conserved in other LAAOs, and the mutation of W570 to another residue may inactivate AncLAAO-N4. To avoid this negative effect, we attempted to widen the space by mutating residues around W570. Here, movement of W570 appeared to be prohibited by two residues, D249 and Y567 (Fig. 5c). In addition, L246 and Q536 can directly interact with the side chain of L-Val by mutation (Fig. 5c). Based on these predictions, we screened AncLAAO-N4 variants of which these three residues were mutated by following procedures as shown in Supplementary Fig. 9a. Screening of AncLAAO-N4 variants was performed by the colorimetric method (Supplementary Fig. 9b). In the first round of screening, random mutagenesis was introduced into V246 and W570, whereas the Y567F mutation was introduced because the aromatic ring of Y567 would be structurally important for AncLAAO-N4 by forming hydrophobic interactions with other residues. Finally, the AncLAAO-N4(Y567F) variant (N4-DQF) could be obtained in the first round of screening, and three AncLAAO-N4 variants, AncLAAO-N4(D249F/Q536W/Y567F) (N4-FWF), AncLAAO-N4(D249A/Q536G/Y567F) (N4-AGF), and AncLAAO-N4(D249V/Q536L/Y567F) (N4-VLF) could be obtained in the second round of screening (Supplementary Fig. 9a).

Enzyme kinetic analysis of AncLAAO-N4 (N4-DQY) and the variants was performed to show that activity toward L-Val is improved by the mutations. Plots of initial velocity of the variants toward six amino acids and kinetic parameters are shown in Supplementary Fig. 10 and Supplementary Table 7, respectively. Differences in $k_{cat}/K_m$ values of the variants is indicated by the color gradient map indicating that, as expected, we can obtain the variants of which activity toward L-Val is improved over N4-DQY through this screening. In fact, the values toward L-Val clearly increased for the three variants (N4-FWF, N4-AGF, and N4-VLF) compared with N4-DQY (Fig. 5d). The change of enzyme kinetics parameters toward L-Val indicated that $k_{cat}$ and/or $K_m$ values were improved in the three variants (N4-FWF, N4-AGF, and N4-VLF). The $k_{cat}$ values for N4-FWF and N4-VLF were >3-fold higher than that of N4-DQY, and $K_m$ values for N4-AGF and N4-VLF were >3-fold lower than that of N4-DQY (Supplementary Table 7). As expected, the mutation enables W570 to move more flexibly to reduce the steric hindrance when L-Val binds to the active site; and this may improve the parameters. In the variants, the $k_{cat}/K_m$ value of N4-AGF and N4-VLF toward L-Ala and D,L-phenylglycine (D,L-PG) also improved (Fig. 5d). On the other hand, the values toward L-Gln were decreased for the three variants (Fig. 5d). This may be caused by mutating Q536 to other hydrophobic residues so that the hydrogen bond interaction between Q536 and the side chain amide group of L-Gln cannot be formed. FAD contents of AncLAAO-N4 and their variants were estimated by measuring the UV-Vis absorption change at 280 and 450 nm. The contents were distributed ranging from 48% to 56%, suggesting that, although we may underestimate absolute values for $k_{cat}$ and $k_{cat}/K_m$ because the values were calculated utilizing protein concentrations estimated by measuring UV-Vis absorption changes at 280 nm; the difference of the contents in the variants hardly affects the relative comparison of values.

Deracemization of D,L-Val was performed utilizing N4-VLF, which has the highest activity of the variants; the $k_{cat}/K_m$ values of this variant were more than 12-fold higher than that of N4-DQY (Supplementary Table 7). After the reaction was carried out for

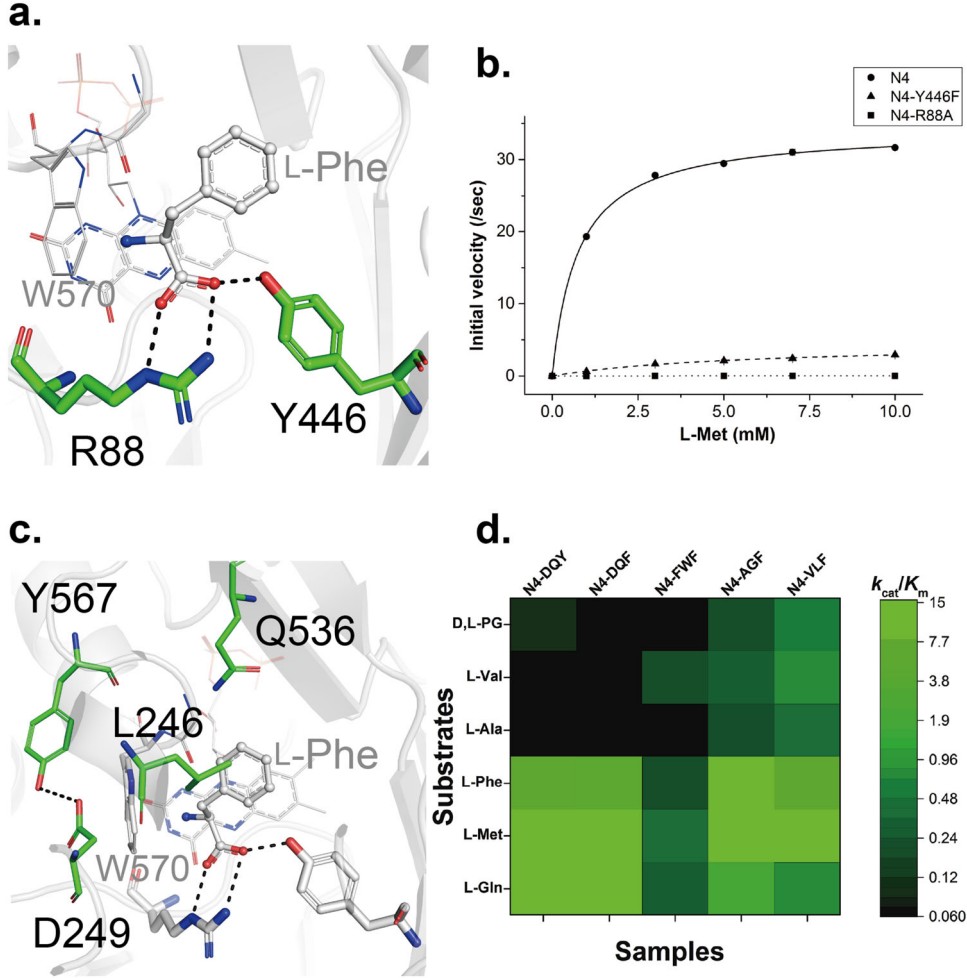

**Fig. 5 Enzyme kinetics analysis of loss-of-function and gain-of-function variants of AncLAAO-N4.** Active site residues of AncLAAO-N4 which form hydrogen bond interactions with the main chain carboxyl group of substrates (**a**) and enzyme kinetic plots of AncLAAO-N4 and the variants (Y446F and R88A) (**b**). Active site residues of AncLAAO-N4 could be predicted from combinational analysis of multiple sequence alignment of the designed AncLAAOs (Supplementary Fig. 1) and crystal structure of AncLAAO-N5; in fact, AncLAAO-N4 shares high sequence identity with AncLAAO-N5 (about 73%, Supplementary Table 1), and active site residues are highly conserved to each other (* and upward triangle in Supplementary Fig. 1). Active site residues of AncLAAO-N4 which were mutated to other residues to improve activity toward L-Val (**c**). Difference of $k_{cat}/K_m$ values of AncLAAO-N4 and variants indicated by a color gradient (**d**). The samples which are utilized in the measurement were as follows: N4-DQY (AncLAAO-N4, which is represented as N4), N4-DQF (N4-Y567F), N4-FWF (N4-D249F/Q536W/Y568F), N4-AGF (N4-D249A/Q536G/Y568F), and N4-VLF (N4-D249V/Q536L/Y568F), respectively. Six of the amino acids (L-Gln, L-Met, L-Phe, L-Ala, L-Val, and D,L-phenylglycine [D,L-PG]) were utilized as substrates to determine enzyme kinetic parameters. The parameters are represented in Supplementary Table 7.

**Table 3 Enzyme kinetic parameters of AncLAAO-N4 and variants (R88A and Y446F) toward L-Met[a].**

| Samples | $k_{cat}$ /sec | $K_m$ mM | $k_{cat}/K_m$ /(s*mM) |
|---|---|---|---|
| AncLAAO-N4 | 34.2 ± 0.3 | 0.76 ± 0.04 | 45.0 |
| AncLAAO-N4(Y446F) | 4.5 ± 0.3 | 5.6 ± 0.8 | 0.80 |
| AncLAAO-N4(R88A) | 0.007 ± 0.000 | 1.1 ± 0.2 | 0.006 |

[a]The measurement of enzyme kinetic parameters was performed independently three times (N = 3).

48 h, D-Val could be obtained at 87% yield with high enantioselectivity (98%, *ee*) (Supplementary Fig. 11b, c). N4-VLF can be overexpressed by the BL21(DE3) expression system as well as N4-DQY; more than 200 units/L of N4-VLF can be obtained after purification by Ni-affinity chromatography (Supplementary Table 3). As a limitation of N4-VLF, its thermal

stability was reduced by the mutation; the $t_{1/2}$ value was >20 °C lower than that of N4-DQY (Supplementary Fig. 11a). Thus, the deracemization reaction was performed at 20 °C.

Taken together, we can prove that AncLAAO-N4 has potential as a biocatalyst to synthesize enantiopure D-amino acid derivatives as well as LAADs as previously reported[24]. Type-II LAADs are broadly utilized biocatalysts to synthesize keto acids and D-amino acid derivatives from L-amino acid derivatives. One of the advantages of LAADs as biocatalysts is that they do not generate $H_2O_2$, which induces decarboxylation of the products[24]. Recently, high-throughput screening of LAAD variants could be achieved by detecting generated keto acids with colorimetric methods[37]. The availability of structural information and these screening systems accelerate the application of LAADs[38,39]. On the other hand, the application of LAAOs is a new approach because several microbial LAAOs have recently been successfully overexpressed by an *E.coli* expression system[40,41]. There are several advantages in the application of LAAOs compared with LAADs. LAAOs do not require an external electron transfer

system and chemical reductants to have the reaction proceed[24]; this can reduce the costs of performing reaction. In addition, LAAOs can be purified and applied in enzyme immobilization; this can enhance the stability of the enzymes significantly[41]. High enzymatic activity of LAAOs would also be a merit for biocatalysts. Both LAADs and LAAOs have different capabilities, so the enzyme selected (LAADs or LAAOs) should be based on the needs of the specific synthesis.

## Discussion

Since DAAOs have been used to synthesize various chemicals with chemoenzymatic reactions, the potential of LAAOs for application to this type of reaction has been of interest[22,42,43]. However, this application was hampered because there are no LAAOs which satisfy two key conditions: (a) production of highly stable LAAOs by an *E.coli* expression system and (b) availability for results of structural and functional analysis of LAAOs. In this study, we designed two AncLAAOs, AncLAAO-N4 and AncLAAO-N5, which satisfy these conditions; AncLAAO-N4 is a highly stable LAAO which can be expressed in a heterologous expression system, and AncLAAO-N5 provides substrate recognition and the reaction mechanism of AncLAAOs at the molecular level. Summarizing the results obtained in this study, we were able to design AncLAAO-N4 variants in which substrate selectivity toward L-Val can be improved by the combination of a structure-based rational design method, and a high-throughput screening method which was adopted to screen useful variants for DAAOs[19] and amine oxidases[20,44,45]. Based on this research, we believe that the activity and substrate selectivity of AncLAAOs give them the potential to be key enzymes in various chemoenzymatic reactions[14].

## Methods

**Reconstruction of five ancestral L-amino acid oxidases**. In previous study, we generated a total of five ancestral LAAOs (AncLAAO-N1, AncLAAO-N2, AncLAAO-N3, AncLAAO-N4, and AncLAAO-N5), and only AncLAAO-N1 was expressed as AncLAAO[27]; the protein and nucleotide sequences of the AncLAAOs are listed in Supplementary Data 1 and 2, respectively. In this study, we attempted to express the remaining four AncLAAOs. Results of multiple sequence alignment and phylogenetic analysis of the AncLAAOs and ancestral L-arginine oxidase (AncAROD) are represented in Supplementary Figs. 1 and 2, respectively. The sequence identity among AncLAAOs and AncAROD is shown in Supplementary Table 1.

**Overexpression and purification of AncLAAOs**. Five genes encoding each of the AncLAAOs were synthesized and cloned into the pET28a vector via the *Nco*I and *Xho*I sites by ordering GeneScript. The plasmids were transformed into the BL21 (DE3) strain. The strain was cultivated in 1 l of LB broth at 37 °C. The temperature was decreased to 20 °C when the $OD_{600}$ reached 0.5–0.8, and then IPTG (isopropyl-β-D-thiogalactopyranoside) was added to a final concentration of 0.5 mM. After the cells were grown for approximately 15 h, the cells were collected by centrifugation. The cells were suspended into buffer A (20 mM Tris-HCl [8.0] and 10 mM NaCl). After sonication of the cells, the supernatant was collected by centrifugation at 11,000 *g* for 40 min. The supernatant was applied to $Ni^{2+}$-sepharose 6 fast flow column (GE Healthcare, Uppsala, Sweden), and the column was washed with 30 mL of buffer A. The samples were eluted utilizing 15 mL of buffer A containing 10, 40, 70, 100, and 300 mM imidazole. The samples containing AncLAAOs (mainly fractions containing 70 and 100 mM imidazole) were utilized for further purification. The eluted samples were concentrated and applied to the Superdex 200 pg column (GE Healthcare, Uppsala, Sweden) equilibrated by buffer A. Purity of AncLAAO was confirmed by SDS-PAGE (Supplementary Fig. 3a). Protein concentration of the purified samples was estimated by measuring the absorption change at 280 nm by UV-Vis spectrometer. The purified AncLAAO was utilized in the following biochemical assay. FAD contents of AncLAAO-N4 and the variants were estimated utilizing the following equation.

$$FAD contents = \frac{Abs(280nm)/Abs(450nm)}{9.65} \times 100$$

Here, the denominator value was calculated by dividing molar absorption coefficient value at 280 nm of AncLAAO-N4 ($\mathcal{E}_{280} = 109,000$ M/cm), by the value at 450 nm of FAD ($\mathcal{E}_{450} = 11,300$ M/cm). Abs (280 nm) and Abs (450 nm) were represented as UV-Vis absorption value at 280 and 450 nm region of each samples, respectively.

**Analysis of substrate selectivity, thermal stability, and long-term stability of AncLAAOs**. AncLAAO-N4 activity was measured by quantifying the amount of $H_2O_2$ produced by progress of the enzymatic reaction. The assay buffer was composed of the following reagents: 100 mM Bis-tris-HCl (pH 7.0), 1.5 mM 4-aminoantipyrine, 2.0 mM phenol, 50 U/mL peroxidase, and 10 mM substrates. Here, 20 L-amino acids and D-amino acids were utilized as substrates. By adding purified AncLAAO-N4 to the assay buffer, the reaction was started. Utilizing a UV-Vis spectrometer (UV-2450, Shimazu), the initial velocity of AncLAAO-N4 was estimated by monitoring time-dependent UV-Vis spectrum change at 505 nm derived from *N*-ethyl-*N*-(2-hydroxy-3-sulfopropyl)aniline. The produced pigment bears $\mathcal{E}_{505} = 12,700$ M/cm[46]. The relative activity toward 20 L-amino acids was calculated by defining L-Gln (AncLAAO-N4) and L-Met [AncLAAO-N1 and AncLAAO-N4(D249V/Q536L/Y568F)] activity to 100% (Supplementary Table 3).

Thermal stability of AncLAAO-N1, N4, and N5 was determined as follows. The distilled enzyme solution was incubated at 30–80 °C for 10 min without substrate. The heat-treatment samples were moved to an ice-bath. The remaining activity was measured by applying the identical procedure to that of the measurement of substrate selectivity (Fig. 2a). To measure long-term stability, the enzyme solution was incubated at 30 °C for 1–168 h without substrate. After the heat-treatment process, the same procedure was adopted to measure remaining activity (Fig. 2b).

**Deracemization and stereoinversion of D,L- and L-amino acid derivatives to D-forms with preparative scale**. Deracemization and stereoinversion of D,L- and L-amino acid derivatives (D,L-**1a**–**1e**, and L-**1f**–**1h**) was performed as follows. A total 100 mL of reaction buffers (100 mM KPB [pH 7.0], 150 mM $NH_3$:$BH_3$, 200 U/mL Catalase, and 10 mM substrates [D,L-**1a**-**1e**, and L-**1f**-**1h**]) were first prepared. The deracemization and stereoinversion reaction was started by adding a total of 3 mg (D,L-**1a** and **1b**) and 2 mg (D,L-**1c**–**1e** and L-**1f**–**1h**) of purified AncLAAO-N4, respectively. The reaction was progressed for 12–48 h at 20 °C. After the reaction, the identical procedure described in the previous study[27] was adopted to end the reaction. The time course of deracemization of D,L-**1a** (Fig. 2c) and D,L-**1b** (Fig. 2d) was determined with reference to the previous study[27]. For the characterization of the products, please see the Supplementary Fig. 12. For the compounds D-**1b** and **1e**, products were isolated by applying the identical procedure reported by Parmeggiani et al.[4].

**HPLC analyses**. Reverse phase HPLC analyses were performed to evaluate deracemization of racemic amino acids. Shimadzu apparatus (Prominence) connected to UV-Vis detector (SPD-20AV, Shimadzu) and CROWNPAK CR-I(+) column (length/internal diameter = 150/3.0 mm; DAICEL, Osaka, Japan) was utilized in this study. Elution of the compounds was monitored by measuring UV spectra change at 210 nm. Mobile phase composition, retention time, and flow rate for each condition were set as shown in Supplementary Table 4. The HPLC chromatograms for D,L-**1a**-**1b**, D,L-**1c**–**1e**, and D- or L-**1f**–**1h** are shown in Supplementary Figs. 4, S5, and S6, respectively. The conversion rate and enantio excess [*ee* (%)] were calculated by the following equation:

$$ee(\%) = [(D_{area}) - (L_{area})]/[(D_{area}) + (L_{area})] \times 100$$

$$Conversion\ rate(\%) = (D_{area})_{after\ reaction}/[(D_{area}) + (L_{area})]_{before\ reaction}$$

Here, $D_{area}$ and $L_{area}$ represent the peak area of HPLC corresponding to D- and L-isomers, respectively. The conversion rate and *ee* values are represented in Table 1. Chemical assignment of the product was performed by high resolution mass spectrometry (HRMS) analysis (Supplementary Fig. 11). The analysis was performed using Q Exactive (Thermo Fisher Scientific, MA, USA), equipped with a Heated Electrospray Ionization (HESI-II) probe. Compounds were analyzed by a positive ion mode (the scan range: *m/z* 100–300).

**Crystallization and X-ray data collection of AncLAAO-N5**. The purified AncLAAO-N5 samples were concentrated to approximately 45 mg/mL. Crystallization of AncLAAO-N5 for phase determination was performed by the following procedures. First, the AncLAAO-N5 samples were mixed with the compounds at a ratio of 80 µL (AncLAAO-N5): 10 µL (1.0 M NaI): 5 µL (0.1 M TCEP): 5 µL (30% [w/v] 6-aminohexanoic acid). By mixing 1.5 µL protein solution with 1.0 µL reservoir solution (12% [w/v] PEG4000 and 0.1 M NaOAc [pH 4.6]), iodide binding AncLAAO-N5 crystals were obtained. The crystals were soaked in cryoprotectant solution (10% [w/v] PEG4000, 0.1 M NaOAc [pH 4.6], 1.5% [w/v] 6-aminohexanoic acid, 5 mM TCEP, 0.2 M NaI, and 20% [v/v] ethylene glycol) quickly, and the crystal was flash-cooled under a liquid nitrogen stream (100 K). X-ray diffraction data were collected using a Pilatus3 detector at BL5A in Photon Factory (Tsukuba, Japan). The data were integrated and scaled by XDS[47] and SCALA[48]. The initial phase was determined by the iodide-SAD method. Thirty-nine anomalous sites could be assigned by analyzing the data with AutoSol implemented in PHENIX software[49]. Model building was performed by AutoBuild[49] and COOT[50]. Finally, the initial structure of AncLAAO-N5 could be obtained.

Crystallization of the ligand-free form of AncLAAO-N5 was performed by the following procedure. Prior to performing crystallization, we prepared the protein solution by mixing 95 µL of the concentrated AncLAAO-N5 with 5 µL of 0.1 M

TCEP. Of this solution,1.0 µL was mixed with 1.0 µL of reservoir solution (1.0 M NaCl and 0.1 M citric acid [pH 3.5]), and the ligand-free form of AncLAAO-N5 crystals were obtained. The crystals were quickly soaked in the following cryoprotectant solutions to determine structures of ligand-free (2.4 M NaCl, 0.1 M citric acid [pH 3.5], 20% [v/v] PEG400), L-Gln binding (30% [v/v] PEG400, 0.1 M NaOAc [pH 4.6], 0.1 M MgCl$_2$,10 mM L-Gln), L-Trp binding (30% [v/v] PEG400, 0.1 M NaOAc [pH 4.6], 0.1 M MgCl$_2$, 5 mM L-Trp), and L-Phe (30% [v/v] PEG400, 0.1 M NaOAc [pH 4.6], 0.1 M MgCl$_2$, 5 mM L-Phe) binding forms, respectively. The soaked crystals were flash-cooled and X-ray diffraction data collection was performed by applying the identical procedure to the data collection of the iodide binding AncLAAO-N5 crystals. The initial phase was determined by molecular replacement method with MOLREP[51] software using the initial structure of AncLAAO-N5. Model building and refinement were performed using COOT[50] and either REFMAC[52] or PHENIX[49], respectively. All figures were prepared by PyMOL[53]. Crystallographic parameters were represented in Table 2.

**Site-directed mutagenesis of AncLAAO-N4**. Plasmids with AncLAAO-N4 cloned into pET28a vector were utilized as a template. Site-directed mutagenesis was performed utilizing a QuikChange Lightening Multi site mutagenesis kit (Agilent technologies, Santa Clara, CA). Primers utilized to design variants are listed in Supplementary Table 6. Confirmation of AncLAAO-N5 variants was performed by DNA sequencing.

**Mutation library screening of AncLAAO-N4 to improve activity toward L-Val**. The plasmids of which the 249[th] and 536[th] residues of AncLAAO-N4(Y567F) were mutated randomly were transformed into BL21(DE3) strains. Colonies containing the random mutants were picked up and inoculated into 200 µL LB broth with kanamycin dispensed in 96-well plates. Cells were cultured at 37 °C for 7 h, and at 23 °C for overnight after the addition of 1.0 mM IPTG at the final concentration. After the cells were collected by centrifugation at 1480 $g$ for 30 min, 100 µL of bacterial protein extraction reagent was added and incubated for 30 min at 30 °C. Then, 20 µL of supernatants were transferred to each batch of new 96-well plates containing assay buffer (100 mM Bis-tris-HCL [pH 7.0], 1.5 mM 4-aminoanti-pyrine, 2.0 mM phenol, 50 U/mL peroxidase, and 5 mM L-Val). The schematic view on how to screen variants is described in Supplementary Fig. 9.

**Determination of enzyme kinetic parameters for AncLAAO-N4 and their variants**. Kinetic parameters for six amino acids (L-Met, L-Gln, L-Phe, L-Val, L-Ala, and D,L-phenylglycine) of AncLAAO-N4 and their variants were estimated utilizing the following concentrations of substrates: 3–30 mM L-Gln (Supplementary Fig. 10a); 1–30 mM L-Met (Supplementary Fig. 10b), L-Phe (Supplementary Fig. 10c), L-Ala (Supplementary Fig. 10d), L-Val (Supplementary Fig. 10e); and 1–6.5 mM D,L-phenylglycine (Supplementary Fig. 10f). The procedure to evaluate the initial velocity of enzymes was identical to that for the measurement of substrate selectivity. Kinetic parameters were estimated by fitting the initial velocity to Michaelis–Menten equation with non-linear least square method; the parameters are listed in Supplementary Table 7.

**Reporting summary**. Further information on research design is available in the Nature Research Reporting Summary linked to this article.

## Data availability
Protein and DNA sequence data for AncLAAOs are registered in Supplementary Data 1 and 2, respectively. PDB data for AncLAAO-N5 are available from the PDB database (PDB ID: 7C4K, 7C4L, 7C4M, and 7C4N). Validation reports for the PDB data are available in the Supplementary Data 3–6.

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

## Acknowledgements

X-ray data were collected at the synchrotron facilities of the Photon Factory (PF) using beamline BL-5A (2016G102 and 2018G006). The authors are grateful to the beamline staff for their assistance with the experiments performed at PF and SPring-8. This work was supported by JSPS KAKENHI Grant Numbers 16K18688, 18K14391, 17H06169, and 17K06931.

## Author contributions

S.N. managed this study and wrote the paper. K.K., S.N., Y.M., and Y.K. performed the enzyme kinetics assay. S.N. performed X-ray crystallography. F.H. performed HRMS analysis of the obtained compounds. S.N. and S.I. contributed to the writing of the manuscript.

## Competing interests

The authors declare no competing interests.
