## [Peer Review File · Communications Chemistry]

Reviewers' comments:

Reviewer #1 (Remarks to the Author):

This paper describes the enzymatic properties, crystal structures, D-amino acid production, and protein engineering of ancestral L-amino acid oxidases (AncLAAOs) obtained by ancestral sequence reconstruction. The authors showed that some of AncLAAOs have much higher thermal stability than an AncLAAO previously obtained by the same group and that one of them also efficiently catalyzed the deracemization and the stereoinversion of racemic amino acids and L-amino acids to D-amino acids. In addition, the authors determined the crystal structure of an AncLAAO and showed a substrate-binding mode in AncLAAO. Furthermore, based on the crystal structure, the authors changed the substrate specificity of an AncLAAO by semi-rational mutagenesis. This paper provides novel findings and insights on the enzyme LAAO and would be of interest to people in the field of the fundamentals and applications of amino acid-relevant enzymes. However, I have many concerns to be addressed, as described below.

Comments and suggestions:

1. The authors determined the activity and the enzyme kinetic parameters of AncLAAOs and the variants without the addition of exogenous FAD into the reaction mixture. There is a possibility that their FAD content might be different among them. The difference might affect their catalytic properties and mislead to false results. Therefore, the authors should confirm their FAD content or determine their catalytic properties in the reaction mixture containing exogenous FAD.

2. It is unclear whether AncLAAO-N2, N3, N4, and N5 were obtained in this study or the previous study (Nakano et al. ACS Catal. 2019). If they were obtained in the previous study, please revise the sentences related to their acquisition to clearly understand that through the manuscript, including Materials and Methods.

3. P7, Fig.S3B, and P14. The optimum pH of AncLAAO-N1 and N5 might be different in the text and Fig. S3B. If the optimum pH of AncLAAO-N5 in Fig. S3B is correct, the optimum pH of AncLAAO-N5 is much different from that of AncLAAO-N4 (pH 6.0 and pH7.5, respectively), suggesting that their active site structures might be different. Therefore, the predictions based on AncLAAO-N5 crystal structure could not be applied to AncLAAO-N4.

4. P9. The dimer formation of AncLAAO-N5 should be experimentally determined using gel filtration chromatography.

5. P11. Why was the substrate not degraded by the enzyme in the soaking? Are the enzyme crystals inactive form? Did the authors check the activity of the enzyme crystals?

6. P11. The authors claim that the incomplete conversion of several racemic amino acids may be caused by reaction products, keto acids and carboxylic acids. However, various factors, such as catalytic activity, substrate affinity, hydrolysis and reduction rates of imino acids, might be considered to be also involved in the conversion. Please more carefully discuss the possible reason for the incomplete conversions.

7. P11. AncLAAOs also exhibited higher activity toward the acidic amino acid L-Glu. Are there any basic amino acid residues that can interact with the side chain of L-Glu in the active site? This might be one of the important information on the substrate specificity of AncLAAOs.

8. P11 and P12. "the oxidation of L-Phe could progress with the same mechanism as indicated by other LAAOs" Please added the reference(s) described the reaction mechanism of other LAAOs. In addition, Are there any reports showing the binding of D-amino acids to the active site of LAAOs or the inhibition of LAAO activity by D-amino acids? If not, the binding model of D-amino acids to

LAAO should be removed.

9. P14, Table 2. Based on the crystal structures of AncLAAO-N5 (Fig. 4), R88A mutation of AncLAAO4 is likely to more largely decrease the substrate-binding affinity. The K_m value of R88A mutant was not so different from that of the wild type. Could the authors explain the reason?

10. Fig. 2 and 3. Is there a hydrogen bond between the α -carboxy group of substrate and Y447? If there is, the formation of the hydrogen bond should be added.

11. P15. In addition to the effect of Q536 mutation on the substrate specificity, the possible contribution of Y567F and D249 mutations to the increase in the catalytic properties toward L-Val should be discussed with their effects on K_m and k_{cat} values.

12. Table S5. L-Tyrosine is theoretically not dissolved at a concentration of 10 mM. Please add how to prepare the solution.

13. P3 and P6. "(Fig.1A)" should be "Scheme1" in P3, and please delete "(A)" from the title of Scheme 1 in P6.

14. P4. The previous name of AncLAAO-N1 in their previous paper should also be described.

15. Fig. 1. Protein concentrations should be added in the legend of the evaluation of their thermal stabilities.

16. P7. There are many mistakes in the figure labels. Please correct all of them. Also carefully check other figure and table labels in the text through the manuscript. In addition, "(L-1f and 1h)" should be "(L-1f, 1g, and 1h)".

17. Fig. 4. Please clearly indicate the position of W570.

Reviewer #2 (Remarks to the Author):

This paper reports the characterisation of two L-amino acid oxidases (LAAOs) which have been generated using Ancestral Sequence Reconstruction (ASR). These two new sequences, termed AncLAAO-N4 and AncLAAO-N5, are able to enantioselectively oxidise a reasonably broad range of L-amino acids and hence have been applied in the deracemization of racemic mixtures by combining with a non-selective reducing agent (ammonia-borane). This group previously report a related ASR sequence AncLAAO-N4 (see ref. 26 ACS Catal 2019) but this enzyme suffered from poor stability and thermolability. The new enzymes have $T_{1/2}$ values ca. 25°C higher and hence are more suitable for preparative scale deracemization reactions. The authors have also solved the x-ray structure of the N5 sequence and used this structure to inform mutagenesis of the N4 sequence in order to improve activity towards L-valine.

Although there are a number of papers that describe the use of (engineered) DAAOs and LAAOs for deracemization of amino acids the present manuscript is interesting in that (i) it uses ASR to identify new biocatalysts and (ii) LAAOs are more difficult to work with than DAAOs and importantly give access to D-amino acid products.

There are two important issues that need to be addressed:

1. I could find no evidence of product isolation from any of the preparative scale reactions, only conversions and e.e.'s. Isolated yields should be given for some of the reactions.

2. An important reference is missing which pre-dates reference 19. See Heath et al. ChemCatChem 2014, doi.org/10.1002/cctc.201301008.

Reviewer #3 (Remarks to the Author):

After using this approach successfully in 2019 the authors describe the recombinant expression of two additional L-amino acid oxidases (AncLAAO-N4 and AncLAAO-N5), whose sequences were derived from nodes of a phylogenetic analysis in an ancestral sequence reconstruction. They obtained AncLAAO-N4 with higher stability. They were able to crystallize AncLAAO-N5 and generate X-ray structures for the enzyme without substrate and with L-Gln, L-Phe or L-Trp. Based on this structure they selected three amino acid residues in the active site of AncLAAO-N4 for semi-rational protein engineering to change substrate specificity. They obtained a variant with improved activity towards L-Val and reduced activity towards the two best substrates compared to the wild type enzyme.

The authors present very interesting, novel data. Their experimental approach is convincing.

English language has to be improved throughout the manuscript.

I have two minor remarks:

Page 7, 6 lines from the bottom:

The reaction was completed within 9 (D,L-1a, Fig. 1D) and 12 h (D,L-1b, Fig. 1E),

As there is no Fig. 1E it should read:

The reaction was completed within 9 (D,L-1a, Fig. 1C) and 12 h (D,L-1b, Fig. 1D),

Page 15, middle and Fig. S8:

AncLAAO-N4(Y567F) variant (N4-DQF) could be obtained in the first round of screening. Screening is stated at these two locations. However, there is a primer mentioned for Y567F in Table S9. Therefore, it is unclear to me whether this first mutation was generated by site directed mutagenesis or obtained by screening a mutant collection. The authors should clarify this.

[Response to reviewer 1]

We deeply appreciate the reviewer both for his/ her patience, valuable time and precise comments/ questions. Our answers were written in followings. Modification points were highlighted by red color in the manuscript. The comments/ questions are shown as italic.

1. The authors determined the activity and the enzyme kinetic parameters of AncLAAOs and the variants without the addition of exogenous FAD into the reaction mixture. There is a possibility that their FAD content might be different among them. The difference might affect their catalytic properties and mislead to false results. Therefore, the authors should confirm their FAD content or determine their catalytic properties in the reaction mixture containing exogenous FAD.

As pointed out by this comment, we estimated FAD contents of AncLAAO-N4 and their variants by measuring UV-Vis absorption at 280 and 450 nm. The contents for the variants were as following: 48 (N4-DQY), 51 (N4-DQF), 54 (N4-FWF), 56 (N4-AGA) and 53% (N4-VLF), respectively. This result suggested that, although we may underestimate absolute values of kcat and kcat/Km, difference of the contents among the variants hardly affects to relative comparison of the values as indicated in the manuscript. To complement this point, we added the following sentences in the manuscript.

(After) FAD contents of AncLAAO-N4 and their variants were estimated by measuring the UV-Vis absorption change at 280 nm and 450 nm. The contents were distributed ranging from 48% to 56%, suggesting that, although we may underestimate absolute values for kcat and kcat/Km because the values were calculated utilizing protein concentrations estimated by measuring UV-Vis absorption changes at 280 nm; the difference of the contents in the variants hardly affects the relative comparison of values. (line 16-21, P16)

2. It is unclear whether AncLAAO-N2, N3, N4, and N5 were obtained in this study or the previous study (Nakano et al. ACS Catal. 2019). If they were obtained in the previous study, please revise the sentences related to their acquisition to clearly understand that through the manuscript, including Materials and Methods.

As pointed out by this comment, ancestral LAAOs which located each node on phylogenetic tree were already designed in previous study. To be clear this point, we amended the sentences as followings.

(After)

a. ... we attempted to express a total of five AncLAAOs (AncLAAO-N1, AncLAAO-N2, AncLAAO-N3, AncLAAO-N4, and AncLAAO-N5) which were generated by the ASR method in a previous study (Table S1 for protein sequences and Table S2 for nucleotide sequences)

(27). Here, we successfully expressed AncLAAO-N1, which is called AncLAAO in the previous study (27), and attempted to do the same for the four remaining AncLAAOs. (line 4-9, P6)

b. In previous study, we generated a total of 5 ancestral LAAOs (AncLAAO-N1, AncLAAO-N2, AncLAAO-N3, AncLAAO-N4, and AncLAAO-N5), and only AncLAAO-N1 was expressed as AncLAAO (27); the protein and nucleotide sequences of the AncLAAOs are listed in Table S1 and S2, respectively. In this study, we attempted to express the remaining four AncLAAOs. (line 3-6, P19)

3. P7, Fig.S3B, and P14. *The optimum pH of AncLAAO-N1 and N5 might be different in the text and Fig. S3B. If the optimum pH of AncLAAO-N5 in Fig. S3B is correct, the optimum pH of AncLAAO-N5 is much different from that of AncLAAO-N4 (pH 6.0 and pH7.5, respectively), suggesting that their active site structures might be different. Therefore, the predictions based on AncLAAO-N5 crystal structure could not be applied to AncLAAO-N4.*

To confirm this point, we analyzed conservation of the residues in AnLAAO-N5 which located around 10 Å region of the substrate (L-Gln). Total 66 residues are located in the region, and only R258 is not conserved in the designed AncLAAOs (The 258th residue was Arg in AncLAAO-N5 and Lys in AncLAAO-N1, N2, N3 and N4). The side chain of R258 directed toward solvent exposed region and main chain oxygen atom of R258 located in the region, suggesting that the residual difference may little affect to the substrate recognition and reaction mechanisms. Thus, we expected that prediction of the mechanisms obtained by the analysis of AncLAAO-N5 structure would be helpful to figure out the mechanisms of AncLAAO-N4. To complement this point, we added following sentences in the manuscript.

(After) In fact, there are a total of 66 residues around the 10 Å region of the substrate (L-Gln), and only R258 is not conserved in the five designed AncLAAOs; the main chain carbonyl group of R258 is located in this region. This suggested that, although optimal pH values are different among AncLAAO-N1, N4, and N5 (Fig. S3B), predictions for substrate recognition and reaction mechanisms obtained by the analysis of AncLAAO-N5 structures would be helpful to determine the mechanisms of AncLAAO-N4. (line 3-8, P14)

4. P9. *The dimer formation of AncLAAO-N5 should be experimentally determined using gel filtration chromatography.*

Accordingly, we checked oligomer state of AncLAAO-N5 by gel-filtration chromatography (Fig. S7). To complement this point, we added the following sentence in the manuscript.

(After) AncLAAO-N5 was crystallized utilizing concentrated samples which were collected at the dimer fraction of gel-filtration chromatography (Fig. S7). (line 10-12, P9)

5. P11. *Why was the substrate not degraded by the enzyme in the soaking? Are the enzyme crystals inactive form? Did the authors check the activity of the enzyme crystals?*

By soaking the AncLAAO-N5 crystals into cryoreservoir solution containing substrates (L-Gln, L-Phe and L-Trp), the crystals were immediately bleached, suggesting that FAD in AncLAAO-N5 crystal was completely reduced to FADH. In this situation, the substrates cannot be oxidized by AncLAAO-N5 crystal and bind to their active site. The similar phenomenon was confirmed in another oxidases, and therefore, we predicted that the substrates can bind to AncLAAO-N5 crystal. To complement this point, we added the following description in the manuscript.

(After) Here, AncLAAO-N5 crystal was bleached by soaking, suggesting that FAD in the crystal was completely reduced to FADH. In this situation, AncLAAO-N5 should lose enzymatic activity. There is a large amount of substrate compared with AncLAAO-N5, and therefore, the substrate could bind to the active site as well aid in the structure determination of the substrate binding form of R-amine oxidase (19, 35). (line 4-8, P11).

6. P11. *The authors claim that the incomplete conversion of several racemic amino acids may be caused by reaction products, keto acids and carboxylic acids. However, various factors, such as catalytic activity, substrate affinity, hydrolysis and reduction rates of imino acids, might be considered to be also involved in the conversion. Please more carefully discuss the possible reason for the incomplete conversions.*

Accordingly, we added discussion for the possible reason for the incomplete conversion of several of racemic amino acids as followings.

(After) Here, there are many factors why complete conversion of several racemic amino acids [such as D,L-1a and D,L-1d (Table 1)] to D-isomers could not be achieved, including unsuitability of enzymatic properties of AncLAAO-N4 exemplified by catalytic activity, substrate affinity hydrolysis, and reduction of imino acids. (line 13-16, P11)

7. P11. *AncLAAOs also exhibited higher activity toward the acidic amino acid L-Glu. Are there any basic amino acid residues that can interact with the side chain of L-Glu in the active site? This might be one of the important information on the substrate specificity of AncLAAOs.*

To confirm this point, we analyzed the residues which locate around 5 Å region of substrate (L-Gln). In conclusion, there are two hydrophilic amino acids at the region: R89 and Q537. Structural analysis indicated that R89 forms interaction with main chain carboxyl group of the substrate, and only Q537 can interact with side chain of the

substrate. Based on this result, we predicted that side chain amide group of Q537 is important to recognize L-Gln and L-Glu. To complement this point, we added the following sentence in the manuscript.

(After) Q537 is the only residue which has a side chain amide group placed to form an interaction with the substrate, suggesting that the residue is important for recognizing L-Gln and L-Glu as substrates (Fig. 2C). (line 21-23, P11)

8. P11 and P12. *“the oxidation of L-Phe could progress with the same mechanism as indicated by other LAAOs” Please added the reference(s) described the reaction mechanism of other LAAOs. In addition, Are there any reports showing the binding of D-amino acids to the active site of LAAOs or the inhibition of LAAO activity by D-amino acids? If not, the binding model of D-amino acids to LAAO should be removed.*

Accordingly, we added three references (Pollegioni, L. *et al.*, 2013, *Appl. Microbiol. Biotechnol.*, **97**, 9323-9341; Moustafa I. M. *et al.*, 2006, *J. Mol. Biol.*, **364**, 991-1002; Faust A. *et al.*, 2007, *J. Mol. Biol.*, **367**, 234-248) which describe LAAO reaction mechanism. Currently, we cannot determine D-amino acid binding form of AncLAAO-N4. So to complement this point, we added following changes in the manuscript.

(After) a. ...as indicated by other LAAOs (22, 30, 31),... (line 5 in P12)

b. The model of D-amino acid binding form of AncLAAO-N4 was removed from the Figure (Figure 3).

9. P14, Table 2. *Based on the crystal structures of AncLAAO-N5 (Fig. 4), R88A mutation of AncLAAO4 is likely to more largely decrease the substrate-binding affinity. The Km value of R88A mutant was not so different from that of the wild type. Could the authors explain the reason?*

In R88A variant, K_m value can be approximated to K_d value because of significant decrease of k_{cat} value; the variant may satisfy the conditions of $k_{-1} \gg k_{cat}$ of which k_{-1} and k_{cat} represent substrate dissociation rate and chemical step rate in Michaelis equation, respectively. The comparison of enzyme kinetics parameters between AncLAAO-N4 and R88A variant suggested that R88 is important for catalysis but not for substrate recognition. As one of plausible function of R88, the residue may work to orient the substrate in appropriate position to proceed the catalysis. To complement this point, we added the following description in the manuscript.

(After) This suggested that R88 of AncLAAO-N4 is an essential residue for the catalytic process to occur. Comparable Km values between AncLAAO-N4 and the R88A variant (Table 2) suggested that residues interacting with the side chain of the substrate (L-Met) is

important for substrate recognition. R88 may work to orient the substrate in an appropriate position to allow the reaction to proceed. (line 16-20, P14)

10. Fig. 2 and 3. Is there a hydrogen bond between the α -carboxy group of substrate and Y447? If there is, the formation of the hydrogen bond should be added.

Accordingly, we added hydrogen bond between OH group of Y447 and α -carboxy group of substrate as dotted line. (Figure 2 and 3)

11. P15. In addition to the effect of Q536 mutation on the substrate specificity, the possible contribution of Y567F and D249 mutations to the increase in the catalytic properties toward L-Val should be discussed with their effects on K_m and k_{cat} values.

Accordingly, we added discussion about improvement of enzyme kinetics parameters (k_{cat} and K_m) in the three variants (N4-FWF, N4-AGF and N4-VLF). Based on the improvement, we predicted that the mutation enables W570 to move more flexibly to reduce the steric hindrance when L-Val binds to the active site. To mention these points, we added the following description in the manuscript.

(After) The change of enzyme kinetics parameters toward L-Val indicated that k_{cat} and/or K_m values were improved in the three variants (N4-FWF, N4-AGF, and N4-VLF). The k_{cat} values for N4-FWF and N4-VLF were > 3-fold higher than that of N4-DQY, and K_m values for N4-AGF and N4-VLF were > 3-fold lower than that of N4-DQY (Table S10). As expected, the mutation enables W570 to move more flexibly to reduce the steric hindrance when L-Val binds to the active site; and this may improve the parameters. (line 7-12, P16)

12. Table S5. L-Tyrosine is theoretically not dissolved at a concentration of 10 mM. Please add how to prepare the solution.

Accordingly, we added the sentence that the relative activity toward L- and D-Tyr was measured utilizing substrate saturated solution as following.

(After) The relative activity toward L- and D-Tyr was measured utilizing substrate saturated solution. (caption ^a in Table S5)

13. P3 and P6. "(Fig.1A)" should be "Scheme1" in P3, and please delete "(A)" from the title of Scheme 1 in P6.

Accordingly, we moved Scheme 1 to P3, and notation was amended from Fig.1A to Scheme 1 (P3).

14. P4. *The previous name of AncLAAO-N1 in their previous paper should also be described.*

Accordingly, we added description how we called AncLAAO-N1 in previous study as following.

(After) (AncLAAO-N1 in this study, and AncLAAO in a previous study (27) (line 14, P4).

15. Fig. 1. *Protein concentrations should be added in the legend of the evaluation of their thermal stabilities.*

Accordingly, we added the protein concentration to measure the thermal stabilities as followings.

(After) A total of 1.1, 0.14, and 0.38 μg of AncLAAO-N1, N4, and N5 were utilized in the measurement. (Legends in Figure 1)

16. P7. *There are many mistakes in the figure labels. Please correct all of them. Also carefully check other figure and table labels in the text through the manuscript. In addition, "(L-1f and 1h)" should be "(L-1f, 1g, and 1h)".*

Accordingly, we checked and amended the figure labels in the manuscript. (such as P7)

17. Fig. 4. *Please clearly indicate the position of W570.*

Accordingly, we added label of W570 to indicate their position. (Fig. 4A and C)

[Response to reviewer 2]

We appreciate you to read our manuscript carefully and give kind comments. Our answers were written in followings. Modification points were highlighted by red color in the manuscript. The comments/ questions are shown as italic.

1. I could find no evidence of product isolation from any of the preparative scale reactions, only conversions and e.e.'s. Isolated yields should be given for some of the reactions.

Accordingly, we determined isolated yields for the compounds **1b** (92%) and **1d** (86%) by applying the identical procedure reported by Parmeggiani et al. (F. Parmeggiani *et al.*, *Adv. Synth. Catal.*, 2016, **358**, 3298-3306). To complement this point, we added following sentences in the manuscript.

(After)

- a. ^d Isolated yield was estimated as follows: 92% (**1b**) and 86% (**1e**). (Caption in Table 1)
- b. For the compounds D-**1b** and **1e**, products were isolated by applying the identical procedure reported by Parmeggiani *et al.* (4). (line 4-6, P21)

2. An important reference is missing which pre-dates reference 19. See Heath et al. ChemCatChem 2014, doi.org/10.1002/cctc.201301008.

Accordingly, we cited the suggested reference (20) in the manuscript.

- (After)
- a. ...detecting H₂O₂ production with colorimetric methods (19, 20). (line 17, P3)
 - b. ...and amine oxidases (20, 44, 45). (line 12, P18)

[Response to reviewer 3]

We appreciate you to read our manuscript carefully and give kind comments. Our answers were written in followings. Modification points were highlighted by red color in the manuscript. The manuscript was checked by native English speaker. The comments/questions are shown as italic.

Page 7, 6 lines from the bottom:

The reaction was completed within 9 (D,L-1a, Fig. 1D) and 12 h (D,L-1b, Fig. 1E),

As there is no Fig. 1E it should read:

The reaction was completed within 9 (D,L-1a, Fig. 1C) and 12 h (D,L-1b, Fig. 1D),

Accordingly, we amended the typo as following.

(After) The reaction was completed within 9 (D,L-1a, Fig. 1C) and 12 h (D,L-1b, Fig. 1D), respectively, (line 15, P7)

Page 15, middle and Fig. S8:

AncLAAO-N4(Y567F) variant (N4-DQF) could be obtained in the first round of screening

Screening is stated at these two locations. However, there is a primer mentioned for Y567F in Table S9. Therefore, it is unclear to me whether this first mutation was generated by site directed mutagenesis or obtained by screening a mutant collection. The authors should clarify this.

At 1st round screening, we introduced random mutation at V246 and W570, and site-directed mutation at Y567 to Phe. This is because aromatic ring of Y567 appeared to be structurally important for AncLAAO-N4 by forming hydrophobic interactions with other residues. Through the screening, we could only obtain AncLAAO-N4(Y567F) variant. Based on this mutant, we performed 2nd round screening as shown in Fig. S8B. To complement this point, we added the following sentence in the manuscript.

(After) In the first round of screening, random mutagenesis was introduced into V246 and W570, whereas the Y567F mutation was introduced because the aromatic ring of Y567 would be structurally important for AncLAAO-N4 by forming hydrophobic interactions with other residues. (line 23-26, P15)

REVIEWERS' COMMENTS:

Reviewer #1 (Remarks to the Author):

The manuscript has been considerably revised and improved based upon the comments of reviewers. The corrections described below would be however needed.

1. Please give the more details (methods) for estimating the FAD contents (p.16, line 16-21) into Materials and Methods. Please be careful that FAD, as well as protein, can significantly absorb 280-nm wavelength light.
2. As suggested previously, the optimum pH of AncLAAO-N1 and N5 is different between that in p.6, lines 21-22 and in Fig. S3b. Please correct the difference.

Reviewer #2 (Remarks to the Author):

I am satisfied that the authors have now responded to the points raised by the reviewers.

Reviewer #3 (Remarks to the Author):

The authors have addressed my questions and I am satisfied with their response.

[Response to Reviewer 1]

We deeply appreciate the reviewer both for his/ her patience, valuable time and precise comments/questions. Modification points were highlighted by red color in the manuscript. The comments/questions are shown as italic.

1. Please give the more details (methods) for estimating the FAD contents (p.16, line 16-21) into Materials and Methods. Please be careful that FAD, as well as protein, can significantly absorb 280-nm wavelength light.

Ans. Accordingly, we added the details how to estimate the FAD contents as following.

(After) FAD contents of AncLAAO-N4 and the variants were estimated utilizing following equation.

$$FAD\ contents = \frac{Abs(280nm)}{9.65 \times Abs(450nm)} \times 100$$

Here, the denominator value was calculated by dividing molar absorption coefficient value at 280 nm of AncLAAO-N4 ($\epsilon_{280} = 109,000\ M^{-1}\cdot cm^{-1}$) by the value at 450 nm of FAD ($\epsilon_{450} = 11,300\ M^{-1}\cdot cm^{-1}$). Abs(280nm) and Abs(450nm) were represented as UV-Vis absorption value at 280 and 450 nm region of each samples, respectively. (line 2-8, P17)

2. As suggested previously, the optimum pH of AncLAAO-N1 and N5 is different between that in p.6, lines 21-22 and in Fig. S3b. Please correct the difference.

Ans. Accordingly, we corrected the optimal pH value as following.

(After) Optimal pH values of AncLAAO-N1, N4, and N5 were 7.0, 7.5, and 6.0 (line 24, P5).